Elasmobranch-associated microbiota: a scientometric literature review

Correia Costa Ivana 1 ivanacorreia72@gmail.com
Amorim de Oliveira Mariene 2
Wosnick Natascha 3
http://orcid.org/0000-0002-9451-471X Ann Hauser-Davis Rachel 4
http://orcid.org/0000-0002-0124-8070 Siciliano Salvatore 5
Nunes Jorge Luiz Silva 1
1 Laboratório de Organismos Aquáticos, Universidade Federal do Maranhão , São Luís, Maranhão , Brazil
2 Laboratório de Genética e Biologia Molecular, Universidade Federal do Maranhão , São Luís, Maranhão , Brazil
3 Departamento de Zoologia, Universidade Federal do Paraná , Curitiba, Paraná , Brazil
4 Laboratório de Avaliação e Promoção da Saúde Ambiental, Instituto Oswaldo Cruz , Rio de Janeiro, Rio de Janeiro , Brazil
5 Departamento de Ciências Biológicas, Escola Nacional de Saúde Pública/FIOCRUZ , Rio de Janeiro , Brazil
Rahman Mohammad Shamsur
Electronic publication date: 2022 Nov 2
Publication date: 2022
Volume: 10
Electronic Location ID: e14255
Received 2022 Feb 7; Accepted 2022 Sep 26
Copyright: © 2022 Correia Costa et al.
Copyright year: 2022
Copyright holder: Correia Costa et al.
License: This is an open access article distributed under the terms of the Creative Commons Attribution License, which permits unrestricted use, distribution, reproduction and adaptation in any medium and for any purpose provided that it is properly attributed. For attribution, the original author(s), title, publication source (PeerJ) and either DOI or URL of the article must be cited.
License URL: https://creativecommons.org/licenses/by/4.0/

Keywords: Bacteria, Fungi, Sharks, Batoids

Funding: Coordination for the Improvement of Higher Education Personnel (CAPES–code 001) Protection of Research and Scientific and Technological Development of Maranhão FAPEMA BM-01149/20, FAPEMA—BEPP-03654/15, FAPEMA—BEPP-02106/18 Financial support was supplied to Ivana Correia Costa and Jorge Luiz Silva Nunes through the Foundation for the Protection of Research and Scientific and Technological Development of Maranhão (FAPEMA BM-01149/20, FAPEMA—BEPP-03654/15, and FAPEMA—BEPP-02106/18). Funding was also provided by the Coordination for the Improvement of Higher Education Personnel (CAPES—code 001). The funders had no role in study design, data collection and analysis, decision to publish, or preparation of the manuscript.

==============================
Elasmobranchs provide greatly relevant ecosystem services for the balance of the environments in which they are inserted. In recent decades, sharp population declines have been reported for many species in different regions worldwide, making this taxonomic group currently one of the most threatened with extinction. This scenario is almost entirely due to excessive fishing pressure, but any contributing factor that may cause additional mortality to populations must be mapped and monitored. In a fast-changing world, emerging marine pollution associated with climate change display the potential to increase the spread of infectious agents. These can, in turn, lead to mortality events, both directly and indirectly, by reducing immune responses and the physical and nutritional condition of affected individuals. In this context, the present study aimed to analyze data concerning elasmobranch-associated microbiota, identifying study trends and knowledge gaps in order to direct future studies on this topic of growing relevance for the health of wild populations, as well as individuals maintained in captivity, considering the zoonotic potential of these microorganisms.

Introduction

Many elasmobranch (shark and ray) species have suffered global population declines in recent years, with overfishing identified as the main factor (Pacoureau et al., 2021). Other elements, however, like different environmental stresses, also act as catalysts towards the diversity crisis currently observed for this taxonomic group, including habitat degradation, pollution, and the effects of climate change, such as rising ocean temperatures, acidification, and eutrophication (Sehnal et al., 2021). These can, in turn, alter vertebrate-associated microbiota interactions and lead to structural and functional changes across entire microbiome communities, ultimately affecting host health, increased infectious diseases (Egan & Gardiner, 2016; Nakatsuji et al., 2017) and decreasing the welfare of many aquatic species, including elasmobranchs (Ward & Lafferty, 2004; Pogoreutz et al., 2019).

Marine pollution is most severe along coastlines and in bays, ports and estuaries, due to high wastewater, industrial, agricultural runoff and riverine pollution discharges (Landrigan et al., 2020). Increased ocean pollution, in turn, leads to greater abundance and expansion of the geographical extent of both naturally occurring and human-introduced marine and estuarine pathogens, such as bacteria, toxin-producing algae, viruses, fungi and protozoa (Escobar et al., 2015; Landrigan et al., 2020), while also favoring antibiotic resistance (Nogales et al., 2011). Resistance traits in fact tend to spread quickly among microorganism populations, making infections more difficult to treat (Zhang et al., 2011), comprising another contributing factor to decreasing wildlife populations (Nogales et al., 2011).

Diverse microbial communities, usually formed by bacteria, fungi and viruses (Doane et al., 2017), have been reported in association to elasmobranchs (Bang et al., 2018; Perry et al., 2021). These microbiome components exhibit varied abundances over space and time in response to both ecological host relationships and environmental restrictions (Pogoreutz et al., 2019; Perry et al., 2021). Positive ecological host-microbiome relationships co-evolve naturally, and associated microbiomes can, for example, facilitate nutrient absorption, regulate host metabolism and act against pathogen invasion (Rosenberg & Zilber-Rosenberg, 2018; Wilkins et al., 2019; Perry et al., 2021). However, negative aspects may also emerge, resulting in compromised host health (Doane et al., 2017), as altered host-microbiome associations may benefit the emergence of bacterial and fungal diseases (Pogoreutz et al., 2019).

Elasmobranch-associated microorganisms can cover the host epidermis or occupy enteric cavities and/or microvilli (Rosenberg & Zilber-Rosenberg, 2018; Perry et al., 2021). Autochthonous microbiota have been, for example, associated to different anatomical body areas in sharks and rays, such as the oropharyngeal cavity, integument (Unger et al., 2014; Florio et al., 2016) and several visceral organs (i.e., intestine, liver, spleen, kidney, heart and pancreas) (Marancik et al., 2011; Camus et al., 2013; Camus et al., 2016). Studies concerning infectious diseases in elasmobranchs, however, are not widespread and usually concern pathologies caused by bacteria and fungi only. For example, Vibrio sp. bacteria are often reported for both sharks and rays (Crow, Brock & Kaiser, 1995; Mylniczenko et al., 2007; Tao, Bullard & Arias, 2014), while the fungus Fusarium solani has been pointed out as the highest cause of systemic mycosis in this taxonomic group (Crow, Brock & Kaiser, 1995; Fernando et al., 2015; Desoubeaux et al., 2017). In this regard, the ecological roles (mandatory pathogen, opportunistic pathogen, benign commensals, symbionts) of many elasmobranch-associated microorganisms are still unknown and further microbiota composition assessments are paramount (Mylniczenko et al., 2007; Tao, Bullard & Arias, 2014), becoming even more vital in the face of environmental changes and elasmobranch host vulnerability (Givens et al., 2015; Doane et al., 2017; Ritchie et al., 2017). In this context, the present study aims to carry out a scientometric assessment of studies reporting elasmobranch-associated microbiomes serving as a database for future investigations, focusing on the causative agents of infectious diseases.

Survey methodology

This study comprises an integrative scientometric review following bibliographic searches focusing on negative host interactions in elasmobranchs at the Scopus (Elsevier), Google Scholar (Google) and Pubmed databases. Studies performed on any elasmobranch species, both in captivity and under free-living conditions, were considered.

The keywords comprised “elasmobranchii”, “shark”, “ray”, “microbiome”, “bacteria” and “fungi”, performed by crossing these descriptors using the Boolean operators “OR” and “AND”. Inclusion criteria considered only white literature (articles and scientific notes) published between 1990 and 2021, in both Portuguese and English. Exclusion criteria consisted in gray literature (monographs, dissertations, theses, books, chapters, studies published in event proceedings) and review articles, as well as articles addressing elasmobranch viruses and protozoa.

Results

A total of 54 publications on elasmobranch-associated microbiota studies were found, corresponding to 38 white literature titles, 31 scientific articles on bacterial microbiota, six exclusively addressing fungi as associated microbiota and only one addressing both bacteria and fungi as microbiota components. The search results are presented in Tables 1 and 2, alongside data on each investigated microbiota taxa, elasmobranch host species, sampled host body region and authors.

Table 1 Elasmobranch-associated microbiota studies published between 1990 and 2021 focusing on fungi.

Fungus (taxon)	Host elasmobranch species	Host body area	Reference	
Fusarium solani	Sphyrna lewini	Head and lateral line	Crow, Brock & Kaiser (1995)	
Dasyatispora levantinae	Dasyatis pastinaca	Skeletal musculature	Diamant et al. (2010)	
Paecilomyces lilacinusa, Mucor circinelloidesb, Exophiala pisciphilab*	Sphyrna mokarran, Stegostoma fasciatum	Liver, heart, kidney, spleen and gills	Marancik et al. (2011)	
Fusarium solani	Taeniura melanopsilac, Sphyrna lewinid	Ventral pectoral fin, head and lateral line	Fernando et al. (2015)	
Fusarium solani	Sphyrna lewini	Head and lateral line	Pirarat et al. (2016)	
Exophiala sp.	Cephaloscyllium ventriosum	Head	Erlacher-Reid et al. (2016)	
Fusarium keratoplasticum,
Fusarium solani
Metarhizium robertsii	Sphyrna lewini,
Sphyrna tiburo	Head and lateral line	Desoubeaux et al. (2017)	
Notes:

a Sphyrna mokarran (Liver, heart and gills).

b,b* Stegostoma fasciatum (Liver, kidney, spleen and gills).

c Ventral pectoral fin.

d Head and lateral line.

Table 2 Elasmobranch-associated microbiota studies published between 1990 and 2021 focusing on bacteria.

Bacteria (taxon)	Host elasmobranch species	Host body area	Reference	
Vibrio alginolyticus, V. damsela, V. parahaemolyticus	Carcharhinus plumbeusa, Negaprion brevirostrisb, Carcharhinus limbatusc, Carcharhinus brevipinnad, Carcharhinus leucase, Ginglymostoma cirratumf, Raja eglanteriag, Dasyatis americanah	Intestine, teeth, gills, spine	Buck (1990)	
Aeromonas salmonicida	Carcharhinus melanopterus	Fins, gills, intestine, liver and kidneys	Briones et al. (1998)	
Vibrio spp.	Sphyrna lewini	Cephalic canals and lateral line	Crow, Brock & Kaiser (1995)	
Photobacterium damselal,
Staphylococcus epidermidisj,
Vibrio alginolyticusk	Carcharhinus melanopterus, Triaenodon obesus, Himantura granulata, Carcharhinus limbatus, Orectolobus japonicus, Carcharhinus acronotus, Carcharhinus plumbeus, Cephaloscyllium ventriosum, Chiloscyllium plagiosum, Triakis semifasciata	Blood	Mylniczenko et al. (2007)	
Enterobacter cloacae, Enterobacter aerogenes, Citrobacter freundii, Citrobacter koseri, Proteus mirabilis, Moellerella wisconcensis, Providencia alcalifaciens, Escherichia coli, Citrobacter farmeri, Proteus vulgaris, Leclercia adecarboxylata, Staphylococcus epidermidis, Staphylococcus sciuri, Staphylococcus warneri, Streptococcus, Enterococcus, Staphylococcus hominis, Staphylococcus xylosus	Carcharhinus leucasl, Galeocerdo cuvierm	Oral cavity	Interaminense et al. (2010)	
Mycobacterium avium	Hemiscyllium ocellatum	Oral cavity, clasper, liver, spleen and intestine	Janse & Kik (2012)	
Mycobacterium chelonae	Rhinobatos lentiginosus	Spleen and skin	Anderson et al. (2012)	
Carnobacterium maltaromaticum	Lamna ditropis	Brain, blood, liver and heart	Schaffer et al. (2012)	
Mycobacterium chelonae	Urobatis jamaicensis	Dorsal face and spiracle	Clarke et al. (2013)	
Serratia marcescens	Sphyrna tiburo	Blood, skin, liver, kidney, spleen and brain	Camus et al. (2013)	
Saccharicrinis carchari, Saccharicrinis fermentans	Cetorhinus maximus	Gills	Liu et al. (2014)	
Vibrio spp., Staphylococcus spp., Pasteurella spp.	Carcharhinus limbatus	Oral cavity	Unger et al. (2014)	
Vibrio spp., Pseudoalteromonas spp., Arenibacter spp.,
Nautella spp., Amphritea spp., Shewanella spp.	Narcine bancroftii	Blood	Tao, Bullard & Arias (2014)	
Bacillus amyloliquefaciens	Centroscyllium fabricii	Intestine	Bindiya et al. (2015)	
Proteobacteria, Firmicutes, Actinobacteria, Fusobacteria	Carcharhinus brevipinnan, Rhizoprionodon terraenovaeo, Carcharhinus plumbeusp	Intestine	Givens et al. (2015)	
Edwardsiella piscicida	Taeniura meyeni	Heart, intestine, kidney, liver and spleen	Camus et al. (2016)	
Tenacibaculum maritimum	Carcharias taurus	Skin	Florio et al. (2016)	
Pistricoccus aurantiacus	Cetorhinus maximus	Gills	Xu et al. (2016)	
Burkholderiales, Flavobacteriales, Pseudomonadales	Rhinoptera bonasus	Skin	Kearns, Bowen & Tlusty (2017)	
Microbacterium sp., Stenotrophomonas sp., Pseudomonas stutzeri, Pseudomonas putida, Psychrobacter pacificensis, Bacillus cereus, Pseudomonas sp., Photobacterium damselae, Vibrio harveyi, Photobacterium sp., Vibrio sp., Pseudoalteromonas sp., Alteromonas sp., Exiguobacterium sp., Bacillus sp., Lysinibacillus sp., Halomonas sp., Bacillus megaterium, Psychrobacter celer, Psychrobacter sp., Marinobacter hydrocarbonoclasticus, Shewanella sp., Marinobacter sp., Vibrio maritimus, Vibrio parahaemolyticus, Paracoccus sp., Exiguobacterium sp.	Rhinoptera bonasusq, Mobula hypostomar, Hypanus sabinuss	Epidermal Mucus	Ritchie et al. (2017)	
Mycobacterium chelonae	Rhinobatos lentiginosus	Gills, blood, spleen, heart, rectal gland and the mesentery	Tuxbury et al. (2017)	
Brucella sp.	Taeniura lymma	Gills	Eisenberg et al. (2017)	
Pseudoalteromonas spp., Erythrobacter spp.,
Limnobacter spp., Idiomarina spp., Marinobacter spp.	Alopias vulpinus	Skin	Doane et al. (2017)	
Acinetobacter sp., Alteromonas sp., Corynebacterium sp., Pseudonocardia sp., Leeuwenhoekiella sp., Mycobacterium sp., Pseudomonas sp., Talassobacillus sp.	Centroscyllium fabricii	Intestine	Johny et al. (2018)	
Enterobacteraceae, Vibrionaceae, Aeromonadaceae, Moraxellaceae, Bradyrhizobiaceae, Pseudomonadaceae, Rhodobacteraceae, Staphylococcaceae e Streptococcaceae	Sphyrna lewini	Intestine	Juste-Poinapen et al. (2019)	
Rhodobacteraceae, Alteromonadaceae, Halomonadaceae	Carcharhinus melanopterus	Skin	Pogoreutz et al. (2019)	
Enterococcus faecalis	Aetobatus narinari	Head	Delaune & Anderson (2020)	
Actinomycetales	Ginglymostoma cirratum, Negaprion brevirostris, Hypanus americanus	Mucus and skin	Caballero et al. (2020)	
Alphaproteobacteria, Gammaproteobacteria	Alopias vulpinus, Rhincodon typus, Triakis semifasciata, Urolophus halleri	Skin	Doane et al. (2020)	
Oceanimonas sp., Acinetobacter sp., Physchrobacter sp., Sediminibacterium sp., Mycobacterium sp., Devosia sp., Cohaesibacter sp., Erythrobacter sp., Ochrobactrum sp., Staphylococcus sp., Corynebacterium sp., Alicyclobacillus sp., Geobacillus sp., Bacillus sp.	Gymnura altavelat, Dasyatis hypostigmau	Skin and stinger	Gonçalves e Silva et al. (2020)	
Haemophilus sp., Vibrio sp., Corynebacterium sp.,
Kordia sp., Salmonella entérica	Ginglymostoma cirratum, Negaprion brevirostri, Carcharhinus plumbeus, Carcharhinus perezii, Galeocerdo cuvier	Cloaca, gills, skin and teeth	Storo et al. (2021)	
Photobacterium damselae, Clostridiaceae, Peptostreptococcaceae, Pseudomonas veronii, Photobacterium sp., Vibrio sp., Mycoplasma sp., Candidatus Heptoplama, Clostridium perfringens, Phyllobacterium sp.	Sphyrna tiburo	Intestine	Leigh, Papastamatiou & German (2021)	
Notes:

a Teeth (V alginolyticus, V parahaemolyticus).

b Teeth (V alginolyticus, V parahaemolyticus).

c Teeth (V alginolyticus, V parahaemolyticus).

d Teeth (V alginolyticus).

e Teeth (V alginolyticus).

f V alginolyticus (gills, intestine, teeth).

g Teeth (V alginolyticus).

h V alginolyticus (spine, teeth).

i Carcharhinus melanopterus, Triaenodon obesus, Himantura granulata, Carcharhinus limbatus, Orectolobus japonicus.

j Carcharhinus melanopterus, Triaenodon obesus, Carcharhinus acronotus, Carcharhinus plumbeus, Cephaloscyllium ventriosum.

k Carcharhinus melanopterus, Himantura granulata, Carcharhinus limbatus, Chiloscyllium plagiosum, Triakis semifasciata.

l Carcharhinus leucas (Citrobacter farmeri, Proteus vulgaris, Leclercia adecarboxylata, Enterobacter cloacae, Citrobacter freundii, Proteus mirabilis, Staphylococcus hominis, Staphylococcus xylosus, Enterococcus).

m Galeocerdo cuvier (Enterobacter cloacae, Enterobacter aerogenes, Citrobacter freundii, Citrobacter koseri, Proteus mirabilis, Moellerella wisconcensis, Providencia alcalifaciens, Escherichia coli, Staphylococcus epidermidis, Staphylococcus sciuri, Staphylococcus warneri, Streptococcus, Enterococcus).

n Carcharhinus brevipinna (Proteobacteria, Firmicutes, Actinobacteria).

o Rhizoprionodon terraenovae (Proteobacteria, Firmicutes, Fusobacteria).

p Carcharhinus plumbeus (Proteobacteria, Firmicutes).

q Rhinoptera bonasus (Exiguobacterium sp. Pseudoalteromonas sp., Bacillus sp., Lysinibacillus sp., Halomonas sp., Vibrio sp., Bacillus cereus, Bacillus megaterium, Psychrobacter celer, Psychrobacter sp., Marinobacter hydrocarbonoclasticus, Alteromonas sp., Shewanella sp, Marinobacter sp., Vibrio maritimus, Vibrio parahaemolyticus, Paracoccus sp. Exiguobacterium sp).

r Mobula hypostoma (Vibrio sp., Pseudoalteromonas sp., Alteromonas sp).

s Hypanus sabinus (Microbacterium sp., Stenotrophomonas sp., Pseudomonas stutzeri, Pseudomonas putida, Psychrobacter pacificensis, Bacillus cereus, Pseudomonas sp., Vibrio harveyi, Photobacterium sp., Vibrio sp.).

t Skin (Oceanimonas sp., Acinetobacter sp., Physchrobacter sp., Mycobacterium sp., Erythrobacter sp.), Stinger (Acinetobacter sp., Sediminibacterium sp., Devosia sp., Cohaesibacter sp., Physchrobacter sp., Erythrobacter sp.).

u Skin (Oceanimonas sp., Acinetobacter sp., Ochrobactrum sp., Staphylococcus sp., Corynebacterium sp., Alicyclobacillus sp., Geobacillus sp.), Stinger (Sediminibacterium sp., Acinetobacter sp., Staphylococcus sp., Corynebacterium sp., Bacillus sp.).

Concerning bacteria, most studies in sharks were carried out in natural environments, representing more than twice the number of studies carried out in captive elasmobranchs from aquaria/oceanaria. The Corynebacterium taxon was reported in six shark species, followed by Haemophilus sp., Vibrio sp, Kordia sp., Salmonella enterica and Staphylococcus epidermidis, present in five of the investigated species (Fig. 1). Studies concerning bacteria in rays, on the other hand, focused on captive specimens from aquaria/oceanaria. The ray species presenting the highest microbiota richness rates comprised the Spiny butterfly ray Gymnura altavela, the Groovebelly stingray Dasyatis hypostigma, and the Caribbean numbfish Narcine bancroftii, and the microbiota taxa Oceanimonas, Acinetobacter, Mycobacterium chelonae and Staphylococcus epidermidis were present in more than one ray species (Fig. 2).

Figure 1 Bacteria isolated from sharks from both aquaria and natural environments.

Abbreviated bacteria names identified in sharks from natural environments and aquaria. Nat, Natural; Aqua, Aquário; Gcuv, Galeocerdo cuvier; Stib, Sphyrna tiburo; Slew, Sphyrna lewini; Cleu, Carcharhinus leucas; Cplu, Carcharhinus plumbeus; Cfab, Centroscyllium fabricii; Nbre, Negaprion brevirostris; Clim, Carcharhinus limbatus; Avul, Alopias vulpinus; Gcir, Ginglymostoma cirratum; Cmel, Carcharhinus melanopterus; Cper, Carcharhinus perezii; Cbre, Carcharhinus brevipinna; Cmax, Cetorhinus maximus; Tsem, Triakis semifasciata; Rter, Rhizoprionodon terraenovae; Rtyp, Rhincodon typus; Tobe, Triaenodon obesus; Ldit, Lamna ditropis; Hoce, Hemiscyllium ocellatum; Cven, Cephaloscyllium ventriosum; Cpla, Chiloscyllium plagiosum; Ctau, Carcharias taurus; Ojap, Orectolobus japonicus; Cacr, Carcharhinus acronotus; Valg, Vibrio alginolyticus; Cory, Corynebacterium sp; Sepi, Staphylococcus epidermidis; Haem, Haemophilus sp.; Kord, Kordia sp.; Sent, Salmonella enterica; Pdam, Photobacterium damsela; Vpar, Vibrio parahaemolyticus; Alph, Alphaproteobacteria; Gamm, Gammaproteobacteria; Rhod, Rhodobacteraceae; Vibri, Vibrio spp.; Acti, Actinomycetales; Eclo, Enterobacter cloacae; Cfre, Citrobacter freundii; Pmir, Proteus mirabilis; Vibr, Vibrionaceae; Ente, Enterobacteraceae; Aero, Aeromonadaceae; Mora, Moraxellaceae; Brad, Bradyrhizobiaceae; Pseu, Pseudomonadaceae; Stap, Staphylococcaceae; Stre, Streptococcaceae; Staph, Staphylococcus spp.; Past, Pasteurella spp; Scar, Saccharicrinis carchari; Sfer, Saccharicrinis fermentans; Paur, Pistricoccus aurantiacus; Bamy, Bacillus amyloliquefaciens; Acin, Acinetobacter; Alte, Alteromonas sp.; Pseud, Pseudonocardia sp; Leeu, Leeuwenhoekiella sp; Myco, Mycobacterium sp.; Pseudo, Psedomonas sp; Tala, Talassobacillus sp.; Pseudoa, Pseudoalteromonas spp.; Eryt, Erythrobacter spp.; Limn, Limnobacter spp.; Idio, Idiomarina spp.; Mari, Marinobacter spp.; Cmal, Carnobacterium maltaromaticum; Eaer, Enterobacter aerogenes; Ckos, Citrobacter koseri; Mwis, Moellerella wisconcensis; Palc, Providencia alcalifaciens; Ecol, Escherichia coli; Ssci, Staphylococcus sciuri; Swar, Staphylococcus warneri; Cfar, Citrobacter farmeri; Pvul, Proteus vulgaris; Lade, Leclercia adecarboxylata; Shom, Staphylococcus hominis; Sxyl, Staphylococcus xylosus; P. veronii, Pseudomonas veronii; Chep, Candidatus Heptoplama; Cper, Clostridium perfringens; Mavi, Mycobacterium avium; Smar, Serratia marcescens; Tmar, Tenacibaculum maritimum; Asal, Aeromonas salmonicida; Alter, Alteromonadaceae; Halo, Halomonadaceae.

Figure 2 Bacteria isolated from batoids from both aquaria and natural environments.

Abbreviated bacteria names identified in batoids from natural environments and aquaria. Nat, Natural; Aqua, Aquaria; Rbon, Rhinoptera bonasus; Galt, Gymnura altavela; Hsab, Hypanus sabinus; Dhyp, Dasyatis hypostigma; Nban, Narcine bancroftii; Mhyp, Mobula hypostoma; Rlen, Rhinobatos lentiginosus; Hgra, Himantura granulata; Hame, Hypanus americanu; Uhal, Urolophus halleri; Ujam, Urobatis jamaicensis; Tmey, Taeniura meyeni; Tlym, Taeniura lymma; Anar, Aetobatus narinari; Regl, Raja eglanteria; Mche, Mycobacterium chelonae; Valg, Vibrio alginolyticus; Ocea, Oceanimonas; Acin, Acinetobacter; Sedi, Sediminibacterium; Bcer, Bacillus cereus; Pdam, Photobacterium; Epis, Edwardsiella piscicida; Bruc, Brucella sp.; Devo, Devosia; Coha, Cohaesibacter; Ochr, Ochrobactrum; Staphy, Staphylococcus; Cory, Corynebacterium; Alic, Alicyclobacillus; Geob, Geobacillus; Baci, Bacillus; Efae, Enterococcus faecalis; Vibri, Vibrio spp.; Pseudoa, Pseudoalteromonas spp.; Aren, Arenibacter spp.; Naut, Nautella spp.; Amph, Amphritea spp.; Shew, Shewanella spp.; Acti, Actinomycetales; Phys, Physchrobacter; Myco, Mycobacterium; Eryt, Erythrobacter; Alph, Alphaproteobacteria; Gamm, Gammaproteobacteria; Bmeg, Bacillus megaterium; Pcel, Psychrobacter celer; Mhyd, Marinobacter hydrocarbonoclasticus; Vmar, Vibrio maritimus; Vpar, Vibrio parahaemolyticus; Pstu, Pseudomonas stutzeri; Pput, Pseudomonas putida; Ppac, Psychrobacter pacificensis; Vhar, Vibrio harveyi.

Considering fungi as microbiota components, most studies were conducted on captive sharks from aquaria/oceanaria, while studies with rays as hosts comprised one assessment for species from natural environments and aquaria/oceanaria. Coincidentally, most of the studied shark species were hammerheads, namely the Scalloped hammerhead Sphyrna lewini, the Smalleye hammerhead S. tudes and the Great hammerhead S. mokarran, associated with three different fungi species each. The fungus Fusarium solani was the most frequent in the analyzed assessments (Fig. 3).

Figure 3 Fungi isolated from elasmobranchs from both aquaria and natural environments.

Abbreviated fungi names identified in elasmobranchs from natural environments and aquaria. Nat, Natural; Aqua, Aquaria; Slev, Sphyrna lewini; Stib, Sphyrna tiburo; Smok, Sphyrna mokarran; Sfas, Stegostoma fasciatum; Cven, Cephaloscyllium ventriosum; Tmel, Taeniura melanopsila; Dpas, Dasyatis pastinaca; Fsol, Fusarium solani; Fker, Fusarium keratoplasticum; Mrob, Metarhizium robertsii; Plil, Paecilomyces lilacinus; Mcir, Mucor circinelloides; Epis, Exophiala pisciphila; Exop, Exophiala sp; Dlev, Dasyatispora levantinae.

Most studies focusing on the elasmobranch-associated microbiota in both natural environments and aquaria were carried out in the United States, comprising 10 scientific articles on free-living specimens and 11 on captive animals, followed by Brazil, India and China with two articles each, all in free-living elasmobranchs from natural environments (Fig. 4).

Figure 4 Countries that conducted studies with free-living animals from natural environments.

The low number of studies on elasmobranch-associated microbiota in the last 30 years indicates a significant knowledge gap, mainly between 90’s and 2007. From 2010, an increasing interest in the subject is noted, increasing 9-fold in 2021 (Fig. 5).

Figure 5 Articles on elasmobranch microbiota published between 1990 and 2021.

Discussion

This review directs the discussion to microorganisms associated with infections in elasmobranchs. Studies on elasmobranch-associated microbiota are scarce when compared to other types of shark and ray assessments. Shark evaluations are more plentiful for animals from natural environments, while studies concerning rays are more frequent for individuals from aquaria/oceanaria.

Fungi belonging to the Fusarium solani species complex (FSSC) are the most prevalent and virulent concerning infections in both humans and animals (Fernando et al., 2015). Infections in the sharks Sphyrna lewini and Sphyrna tiburo caused by Fusarium keratoplasticum, Fusarium solani and Metarhizium robertsii were reported, where the fungi, located on the dorsal and ventral surfaces of the cephalofoil, progressing to the lateral line, cephalic canals, and ampulla of Lorenzini, caused epidermal erosions, ulcers, hemorrhages and white exudates (Desoubeaux et al., 2017). However, Fusarium transmission is still not well understood, especially in aquatic environments, emphasizing the need for further assessments concerning infection by species belonging this genus (Desoubeaux et al., 2017). The authors observed certain injury patterns in the analyzed sharks, such as the location of injuries distributed in the ampullae of Lorenzini and lateral line, suggesting that these may be microorganism gateways. The skin lesions developed in such a way as to spread infection from one individual to another, as the sensory organs present in shark heads are used to probe the environment and contact other individuals (Desoubeaux et al., 2017). In this regard, one study isolated Fusarium solani from the cephalic canal exudate of two of five S. lewini sharks living in an aquarium, comprising the first report of Fusarium solani infection in the lateral line canal system and the third for hammerhead sharks (Crow, Brock & Kaiser, 1995). Lesions were initially observed in the cephalic canals, but extended during months up the lateral canal, leading to granulomatous exudative mycotic dermatitis and resulting in chronic physical and behavioral deterioration, until the specimens required sacrificing. Other studies indicate that the Fusarium genus is associated to significant infections in elasmobranchs, with F. solani responsible for skin lesions characterized by ulcers and hemorrhage of the frontal pectoral fin of a Blotched fantail ray Taeniurops meyeni (also known as Taeniura melanopsila a junior synonym), and also capable of causing white and purulent exudates in the cephalic canals and lateral line, resulting in animal death (Fernando et al., 2015). Cutaneous lesions were also observed in scalloped hammerheads (Sphyrna lewini), characterized by ulcers, hemorrhaging and white and purulent exudates in the cephalofoil cephalic canals and lateral lines (Fernando et al., 2015). Furthermore, these events constituted the first case of fatal infection by members of the Fusarium solani complex (FSSC) in a Taeniurops meyeni stingray and the cause of concomitant infections in scalloped hammerheads (Fernando et al., 2015). In another assessment, a severe fungal infection also caused by F. solani was observed in the cephalic canals and the lateral line system of seven Scalloped hammerhead sharks (S. lewini) in an aquarium in Thailand, leading to extensive and severe necrotizing cellulitis and resulting in animal death (Pirarat et al., 2016). Abnormal clinical signs were observed prior to animal death, such as head shaking, hitting tanks or rocks, swimming on the surface and restlessness, as well as decreased appetite and visible anorexia (Pirarat et al., 2016).

Although the number of analyzed articles on fungi was small, we were able to identify that fungi belonging to the FSSC are extremely dangerous for captive elasmobranchs, resulting in high injury severity, in some leading to death.

Other fungi genera have also been reported as infectious agents in elasmobranchs, including a new species belonging to the microsporidae group, responsible for infecting 30 Common stingrays Dasyatis pastinaca, invading the disc muscles and producing thin and spindle-shaped subcutaneous swellings that developed into massive, elongated, tumor-like lumps, comprising the first record of microsporidium infection in a batoid (Diamant et al., 2010). Two records of progressive systemic mycosis caused by Paecilomyces lilacinus, Mucor circinelloides and Exophiala pisciphila are also available for two captive shark species, the Great hammerhead and the Zebra shark Stegostoma fasciatum, resulting in terminal disease (Marancik et al., 2011). The authors emphasize that these cases, alongside the lack of literature information, reinforce the need for more research and diagnostic samplings to better characterize host-pathogen interactions between elasmobranchs and fungi (Marancik et al., 2011).

In another study, a female Swell shark Cephaloscyllium ventriosum raised in captivity began exhibiting abnormal behavior (swimming in circles and rolling repeatedly), and a macroscopic necropsy and histopathological examination verified cartilage matrix ossification and fibrosis in the skull and cervical vertebrae. The lesions were associated to a deep invasion of the fungus Exophiala sp. (Erlacher-Reid et al., 2016), reinforcing the need to include fungal infections as well as skeletal structure mineralization as a differential diagnosis when evaluating elasmobranchs exhibiting abnormal swimming behaviors.

Bacteria are also responsible for infections in elasmobranchs. One study, for example, reported the development of a large abscess on the dorsal surface of the calvarium and swollen soft tissue around the left spiracle of an adult Yellow stingray Urobatis jamaicensis raised in captivity, identified as associated to mycobacteria. A significant amount of fluid exudate was drained from the site, the specimen was sacrificed and disseminated mycobacteriosis was later confirmed (Clarke et al., 2013). According to the authors, primary mycobacteriosis can lead fish to succumb to opportunistic diseases. Another case of splenic mycobacteriosis was observed in a Freckled guitarfish specimen Pseudobatos lentiginosus raised in captivity, where darkened pigments appeared on the back skin and rostrum erythema, in addition to numerous whitish granulomas of variable size dispersed throughout the splenic parenchyma. The animal died after being transferred to a holding tank (Anderson et al., 2012). Two more cases of mycobacterial infection in the same species, also from aquaria are noted, with Mycobacterium chelonae identified as the responsible agent following histological tissue and blood culture assessments, confirmed by a DNA sequencing analysis after individuals were found dead inside their display tanks (Tuxbury et al., 2017). The authors report that both acute and chronic mycobacteriosis manifestations may occur in this elasmobranch species. An Epaulette shark Hemiscyllium ocellatum specimen raised in captivity also presented granulomas caused by mycobacteria. The specimen stopped feeding and was euthanized (Janse & Kik, 2012). Despite limited sampling, it seems that elasmobranchs maintained in aquariums become susceptible to mycobacterial infections, with a high pathogenic potential noted for this microorganism.

According to Clarke et al. (2013), mycobacterial species are ubiquitous in the environment and have been observed in biofilms from aquaculture systems and drinking water sources. The known routes of infection are by immersion in contaminated water, traumatic inoculation and ingestion of bacteria or infected tissues or protozoa containing the microorganism. However, the mode of transmission of mycobacteriosis in cartilaginous fish species living in aquariums is still poorly understood. Clarke et al. (2013), emphasize that human interactions may comprise an entry route for mycobacteria elasmobranch contamination in captivity.

Concerning other infectious microorganisms, one assessment reported meningitis and/or meningoencephalitis with inflammatory infiltrates observed in specific brain areas in stranded juvenile Salmon sharks Lamna ditropis (Schaffer et al., 2012), comprising the first report of Carnobacterium infection in sharks. The authors emphasize that brain infections caused by this bacterium are a significant cause of morbidity and mortality in juvenile Salmon sharks found stranded along the Pacific coast, specifically in California.

According to the authors, infections can be specific in juvenile salmon sharks due to unknown aspects of their life history, such as coastal migration, behavior or diet, as well as physiological stresses and immune function aspects not shared by adults, emphasizing that investigations into the natural habitat, lifestyle and ecology of juvenile and adult salmon sharks will aid in elucidating the pathogenesis of this disease (Schaffer et al., 2012).

In another report, the bacteria T. maritimum was isolated for the first time in an adult Sand tiger shark Carcharias taurus raised in captivity, the specimen presented skin lesions characterized by the presence of abundant whitish necrotic tissue between the second dorsal fin and the precaudal fossa. After being treated with medication, the specimen fully recovered from the infection (Florio et al., 2016), suggesting that the skin may be a bacteria gateway in sharks. According to the authors, the T. maritimum infection observed in the shark’s skin may have been triggered by mechanical injuries during mating or as a result of aggressive behavior between sharks of the same species (Florio et al., 2016). According to Avendano-Herrera, Toranzo & Magariños (2006), T. maritimum is part of the autochthonous microbial populations of the marine environment, and can be isolated from sediments, tank surfaces and water. This pathogen adheres to fish skin, including the mucus layer, and to extracellular polymeric substances (Avendano-Herrera, Toranzo & Magariños, 2006). Identified as atypical, a new Brucella strain was isolated in another assessment from the gills of a Bluespotted lagoon ray Taeniura lymma raised in captivity that died suddenly during quarantine (Eisenberg et al., 2017). According to the authors, this is the first report of a natural infection by this microorganism in saltwater fish, increasing the host range of this pathogenic genus. Although not enough is known about host-bacterial relationships or possible adaptations, these findings have significantly improved our understanding of the ecology and pathogenic potential of members of the Brucella genus (Eisenberg et al., 2017).

Finally, the first known case of edwardsiellosis in elasmobranchs (the Blotched fantail ray) was reported for Edwardsiella piscicida, where multiple large lesions were noted in the subepicardium and compact myocardium, partially filled with cellular debris and degenerated granulocytes, delimited by variable mixtures of hemorrhage, dispersed lymphocytes and mucin (Camus et al., 2016). According to the authors, much of the knowledge about disease processes in elasmobranchs comes from diagnostic studies carried out in public aquaria. However, although reports of bacterial diseases are limited, this is more likely due to insufficient reporting and diagnostic investigation than to a lack of existing bacterial infections.

Concerning the intestinal elasmobranch microbiome, P. damselae and C. koseri have been confirmed in all tested sharks (Juste-Poinapen et al., 2019), while the characterization of the intestinal microbiome of a free-living Black dogfish Centroscyllium fabricii through a feces analysis revealed a wide variety of bacterial genera. Furthermore, in this case, about 25% of the animal’s gut microbiome was unable to be taxonomically classified at the phylum level, suggesting a high microbial diversity not yet characterized in this microbiome (Johny et al., 2018). In another assessment, the gut microbiota of juvenile Scalloped hammerheads from the Rewa Delta (Republic of Fiji) contained a diverse bacterial community, including members belonging to the Enterobacteraceae, Vibrionaceae, Propionibacteriaceae, Aeromonadaceae, Staphylococcaceae, Streptococcaceae families, which are known as intestinal inhabitants of terrestrial and marine vertebrate species, including humans and many of these microorganisms are considered opportunistic pathogens (Juste-Poinapen et al., 2019). The authors indicate that sewage spillage during the sampling period may be responsible for the presence of some known indicator microorganisms, while dominance variations between bacterial species over time may reflect environmental changes, such as temperature or food and water quality variations.

Regarding the elasmobranch skin microbiome, a microbiological analysis of the epidermal mucus and skin of three elasmobranch species, the Atlantic nurse shark (Ginglymostoma cirratum), the Lemon shark (Negaprion brevirostris) and the Southern stingray (Hypanus americanus) identified a variety of bacterial orders, with the predominance of Actinomycetales (Caballero et al., 2020). Another investigation concerning the skin microbiota of three sharks and a ray also reported a variety of bacterial classes, although with the predominance of Alphaproteobacteria and Gammaproteobacteria (Doane et al., 2020).

The microbiome is a product of both the host and the environment it inhabits and can be affected by environmental variables. Thus, an equilibrium must be achieved between host immune responses and microbial interactions to maintain elasmobranch microbiota community consistency (Doane et al., 2017). For example, Gonçalves e Silva et al. (2020), observed that G. altavela individuals living in natural environments contained specific bacteria and postulated positive health effects due to this microorganism/host interaction. Temperature appears to be the environmental variable most related to the proliferation of infectious agents in marine animals, and abrupt water temperature alterations are a significant source of mortality associated with infections in stranded sharks (Wosnick et al., 2022). In a climate change scenario, this is of particular concern, as an increase in potentially lethal infectious diseases is expected, as well as pathogens associated with sublethal outcomes, such as reduced immune response, physical condition, and fitness which, in turn, can directly affect population recruitment.

Concerning bacteria and fungi, a higher number of investigations concerning captive elasmobranchs is noted compared to animals in natural environments, although a higher microbial diversity has been reported for free-living elasmobranchs. This suggests that marine contamination may be a significant contributor to microorganism diversity, as aquaria are controlled environments without these types of interferences. In fact, high organic matter discharges into coastal ecosystems have become a significant public health issue (Robinson et al., 2016; Fresia et al., 2019), as these effluents contain several contaminants, such as metals, hydrocarbons, pharmaceutically active organic compounds (Bayen et al., 2019) and endocrine disruptors (Santos et al., 2019), in addition to pathogenic microorganisms (Poharkar et al., 2017). In this regard, wastewater can comprise both a reservoir and vehicle for the transmission of pathogenic bacteria and antibiotic resistance mechanisms to aquatic biota, leading to serious consequences for exposed animals, including global declines in fish stocks (Landrigan et al., 2020), and, consequently, to humans, as many contaminated fish species are routinely marketed and consumed. Furthermore, it seems that the bacterial community of rays from the natural environment is complex, with a high diversity of microbiota taxa, some establishing beneficial symbiotic associations and others responsible for diseases in humans and other animals, including fish (Gonçalves e Silva et al., 2020).

Conclusions

The findings reported herein indicate a significant lack of information concerning elasmobranch-associated microbiota, more critical regarding fungi. (i) In this regard, the prevalence of Fusarium solani was observed in the evaluated literature, while the bacteria genera Mycobacterium and Vibrio were the most noteworthy. (ii) The most analyzed elasmobranchs were sharks, with the prevalence of the Scalloped hammerhead Sphyrna lewini. Furthermore, (iii) captive elasmobranchs were more investigated than free-living ones. Moreover, (iv) as diverse microbiota has been reported mostly for a single elasmobranch species, often in a single anatomical area, further studies on the subject are required encompassing other species and body regions, such as the oral cavity, gastrointestinal tract, blood, muscle and gills. In sum, elasmobranch-associated microbiota evaluations comprise a valuable tool concerning elasmobranch health, as this group is susceptible to bacterial and fungal diseases both in the wild and in captivity. However, (v) although concerns have been noted regarding emerging diseases for this ancient group of fish, this subject is still poorly understood, and scarce information on the biodiversity, prevalence and physiological effects of the microbiota associated with cartilaginous fish is available, indicating the need for further investigations in this field of research. As such, (vi) the potential zoonotic of this significant diversity of microorganisms detected in elasmobranchs should be further evaluated in a fast-changing world.

Additional Information and Declarations

Competing Interests

Author Contributions

Animal Ethics

Data Availability

Rachel Ann Hauser-Davis is an Academic Editor for PeerJ.

Ivana Correia Costa conceived and designed the experiments, performed the experiments, analyzed the data, prepared figures and/or tables, authored or reviewed drafts of the article, and approved the final draft.

Mariene Amorim de Oliveira conceived and designed the experiments, performed the experiments, analyzed the data, authored or reviewed drafts of the article, and approved the final draft.

Natascha Wosnick conceived and designed the experiments, analyzed the data, authored or reviewed drafts of the article, and approved the final draft.

Rachel Ann Hauser-Davis conceived and designed the experiments, analyzed the data, authored or reviewed drafts of the article, and approved the final draft.

Salvatore Siciliano conceived and designed the experiments, analyzed the data, authored or reviewed drafts of the article, and approved the final draft.

Jorge Luiz Silva Nunes conceived and designed the experiments, performed the experiments, analyzed the data, prepared figures and/or tables, authored or reviewed drafts of the article, and approved the final draft.

The following information was supplied relating to ethical approvals (i.e., approving body and any reference numbers):

Not applied.

The following information was supplied regarding data availability:

The data are available in the figures and tables.

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
