# Peer review of "Elasmobranch-associated microbiota: a scientometric literature review"

_PeerJ, doi:10.7717/peerj.14255_

## Round 0.1 · original submission · Major Revisions

Though one of the reviewers suggest to reject the manuscript on the ground that Perry et al. 2021 published a very similar article, I am convinced to publish more reviews on Elasmobranchi microbiota. I am requesting the authors to address all the issues raised by the honorable reviewers, especially to address the claim that this manuscript is very similar to the one by Perrry et al.

Reviewer 1 ·

Basic reporting

I have several concerns regarding this review of elasmobranch microbiomes:


1) This field has very recently been reviewed by Perry et al., 2021, and the submitted review is quite similar. The primary difference is a brief discussion of fungi. However, I don't think that this difference is strong enough to warrant a second elasmobranch microbiome review in as many years.
2) I am unclear about the significance of the figures 1, 2 and 3. The text is too small to read and I am not sure about what they mean. Do figures 4 and 5 refer to the total amount of scientific papers? or those published each year? Figure captions need a lot more detail.
3) It is unclear whether the authors used white literature in this study (lines 91-101 are ambiguous).
4) PeerJ states that reviews must be "broad and cross-disciplinary", which does not apply to the submitted review.

Experimental design

Although the study design is adequate, I am concerned that the authors are using white literature (lines 91-101 are unclear). Additionally, much of the document is just paraphrasing the findings of each study, bordering on the line of plagiarism (see lines 218-221 for an example of this). The authors should spend more text developing their arguments and identifying unresolved gaps and less text paraphrasing each study.

Validity of the findings

Please see above.

Reviewer 2 ·

Basic reporting

The manuscript provided by Costa and colleagues is very well laid-out. I found the article easy to read with language and sentence structure sufficient. Relevant and up-to-date references were used in this manuscript. I have a few comments listed below that will help to clarify the manuscript.

Line 138-141: Sentence needs rewriting – ‘during months up’ is confusing
Line 171: ‘(’ needed around Pseudobatos lentiginosus or remove the ‘)’ after the species name
Line 175: should be Mycobacterium chelonae, not Micobacterium
Line 199: Edwardsiellosis
Line 202: Citation needs formatting
The figures are interesting, but require better description to help the reader interpret the data being summarized.

Perry et al (2021) have reviewed much of the microbiome literature regarding sharks and elasmobranchs. This review separates itself by focusing on the identified causative agents of disease that have been reported for elasmobranchs. I think this perspective is important and certainly warrants a review. One thing that may help the reader to understand the intent of the manuscript is to explicitly state how this review separates from Perry et al (2021) as I have mentioned in the previous sentence.

Experimental design

The study design is sufficient. The authors have provided their keywords for searching the literature which will encourage reproducibility if needed.

Validity of the findings

Costa and colleagues have developed an interesting manuscript with several important contributions to the field concerning elasmobranch microbiota. I felt the authors summarized their results clearly.

The conclusion, while nicely summarizing the findings, does leave some unresolved gaps in the moving forwards. The manuscript focused on results from culture base experiments, yet the introduction acknowledges the diverse consortium of the microbes associated with the microbiome. An interesting area of focus would be to bridge the microbiome community patterns in relation to the infectious states. The infectious agent is perhaps a part of the microbiome under normal or homeostatic conditions and then becomes pathogenic when there is some change, whether physiological (host), environmental (temperature or nutrient changes) or just opportunistic. However, at present, we know little about whether the microbiome displays signs of infection prior to or during these infectious events. Building this out would help to strengthen the merit of the manuscript.

Additional comments

No additional comments

---

## Round 0.2 · accepted · Accept

I have invited 21 reviewers to make the manuscript more informative; considering the expert and information gap on this very particular topic of elasmobranch microbiome, more research and review is necessary. Hope this manuscript will contribute a lot to this very specific field of research.

Reviewer 2 ·

Basic reporting

no comment

Experimental design

no comment

Validity of the findings

no comment

Additional comments

The authors have sufficiently addressed the main concern which is the subject matter of the paper and making it distinct from the recently published review that focused on microbiome patterns associated with elasmobranchs.